# Machine Learning for Screening Microvascular Complications in Type 2 Diabetic Patients Using Demographic, Clinical, and Laboratory Profiles

**DOI:** 10.3390/jcm11040903

**Published:** 2022-02-09

**Authors:** Mamunur Rashid, Mohanad Alkhodari, Abdul Mukit, Khawza Iftekhar Uddin Ahmed, Raqibul Mostafa, Sharmin Parveen, Ahsan H. Khandoker

**Affiliations:** 1Department of Electrical and Electronic Engineering, United International University, Dhaka 1212, Bangladesh; mrashid152005@bseee.uiu.ac.bd (M.R.); abdul.mukit64@gmail.com (A.M.); khawza@eee.uiu.ac.bd (K.I.U.A.); rmostafa@eee.uiu.ac.bd (R.M.); 2Healthcare Engineering Innovation Center (HEIC), Department of Biomedical Engineering, Khalifa University, Abu Dhabi 127788, United Arab Emirates; ahsan.khandoker@ku.ac.ae; 3Department of Electrical and Computer Engineering, University of Oklahoma, Tulsa, OK 74135, USA; 4Department of Health Informatics, Bangladesh University of Health Sciences, Dhaka 1216, Bangladesh; sharminparveen@yahoo.com

**Keywords:** microvascular complications, cardiac autonomic neuropathy, diabetic peripheral neuropathy, diabetic nephropathy, diabetic retinopathy, patient profiles, machine learning

## Abstract

Microvascular complications are one of the key causes of mortality among type 2 diabetic patients. This study was sought to investigate the use of a novel machine learning approach for predicting these complications using only the patient demographic, clinical, and laboratory profiles. A total of 96 Bangladeshi participants with type 2 diabetes were recruited during their routine hospital visits. All patient profiles were assessed by using a chi-squared (χ^2^) test to statistically determine the most important markers in predicting three microvascular complications: cardiac autonomic neuropathy (CAN), diabetic peripheral neuropathy (DPN), and diabetic retinopathy (RET). A machine learning approach based on logistic regression, random forest (RF), and support vector machine (SVM) algorithms was then developed to ensure automated clinical testing for microvascular complications in diabetic patients. The highest prediction accuracies were obtained by RF using diastolic blood pressure, albumin–creatinine ratio, and gender for CAN testing (98.67%); microalbuminuria, smoking history, and hemoglobin A1C for DPN testing (67.78%); and hemoglobin A1C, microalbuminuria, and smoking history for RET testing (84.38%). This study suggests machine learning as a promising automated tool for predicting microvascular complications in diabetic patients using their profiles, which could help prevent those patients from further microvascular complications leading to early death.

## 1. Introduction

Diabetes is called a ‘silent killer’ that is killing around 1.6 million people each year, making it the 5th leading cause of death worldwide [1]. There are two types of diabetes, type 1 and type 2. Type 2 is a chronic metabolic disorder and an expanding global health problem in the past decades. It results in hyperglycemia, which reduces the ability of the body’s cells to respond fully to insulin. This situation is called ‘insulin resistance’. In this state, insulin production increases, due to the inaction of the hormone. The global prevalence of type 2 diabetes in low- and middle-income countries was estimated to be 415 million in 2015 and is predicted to rise to 642 million by 2040 [2]. Type 2 diabetes mellitus has been rapidly rising worldwide over the past three decades, particularly in developing countries, including Bangladesh [3]. The prevalence of type 2 diabetes in Bangladesh will be more than 50% within the next 15 years, placing Bangladesh as the country with the 8th largest diabetic population in the world [4]. A study suggests that diabetic prevalence will more than double between 2020 and 2030 [5]. The IDF (International Diabetes Federation) Diabetes Atlas has estimated that if nothing is done, the number of diabetes patients may rise to 629 million in 2045 [6] and cases may double from 151 million [7] from 2000 to 2025 [8]. The prevalence of diabetes is higher in rural areas [9], but it was high for males in urban areas, whereas it was lower in rural areas compared to females in Bangladesh [10,11].

Neuropathies are a common persistent complication of both types of diabetes mellitus that confer morbidity and mortality to diabetic patients. Cardiac autonomic neuropathy (CAN) is associated with an increased risk of mortality [12,13]. A study including 1171 patients with type 1 and type 2 diabetes mellitus using a predefined HRV and spectral analysis of R-R intervals reported abnormal findings for 34.3% of type 2 patients [14]. Neuropathy is the most common microvascular complication of both type 1 and type 2 diabetes mellitus [15,16,17]. A study conducted in the outpatient section of BIRDEM Hospital, Dhaka, Bangladesh found that 19.7% of all registered type 2 patients have diabetic peripheral neuropathy (DPN) [18]. The prevalence of DPN among type 2 diabetic patients is much higher in Europe. A study concludes that 32.1% of the diabetic patients in the United Kingdom, 17.6% in Turkey, and 35.4% in Spain have DPN [19]. The prevalence of DPN increases with the age of the patient and also with the diabetic duration [18,20]. A multi-country study conducted in Asia shows a 58.6% prevalence of micro or macroalbuminuria, indicating an impending pandemic of diabetic renal (i.e., nephropathy) and cardiovascular diseases in Asia [21]. A cross-sectional study with 836 rural Bangladeshi patients showed a high prevalence of retinopathy in Bangladesh [22]. Results from 35 studies from 1980 to 2008 with 22,896 subjects with diabetes showed that the global prevalence for any RET was 34.6% (95% CI 34.5–34.8) [23]. Analyses of the exponential trend revealed an increase in diabetes prevalence among the urban and rural populations at a rate of 0.05% and 0.06% per year, respectively [24]. Increasing age, hypertension, and higher BMI were found to be significant risk factors in the urban and rural communities of Bangladesh [25]. However, the patients with type 2 diabetes in Bangladesh have limited knowledge of its risk factors, cause, and management [26,27]. Depressive diabetic symptoms were found in 29% of males and 30.5% of female participants with diabetes and 6.0% of males and 14.6% of female subjects without diabetes [28].

Most recently, machine learning has emerged in many biomedical applications as a promising tool to aid in decision-making regarding many diseases, including diabetes. In [29], the authors managed to implement a machine learning approach based on decision trees to identify the diabetic patients with or without treatment procedures from their lipid profiles. In addition, Koren et al. [30] developed a trained model capable of diagnosing diabetic patients with drugs that lower blood glucose levels. Moreover, in [30,31], the authors proposed a deep neural network to diagnose diabetic patients from clinical profiles. To recognize patterns among diabetic patients, Alloghani et al. [31] presented several machine learning models that were able of characterizing patients and explain the re-admission procedures. Several other studies [32,33,34,35] utilized machine learning and deep neural networks in many other applications in diabetes diagnostics. However, even though the implementation of machine learning models for diabetes diagnostics showed high levels of performance, there is still a lack of knowledge about its impact on discriminating between the various microvascular complications. In addition, it is essential to be able to determine, both statistically as well as from a machine perspective, which features play a critical role in characterizing these complications in type 2 diabetic patients.

In this paper, a study is conducted to investigate the efficiency of applying a machine-learning-based approach in discriminating between diabetic patients, according to their microvascular complication status (Figure 1). The novelty of the presented approach lies in utilizing only the demographic, laboratory, and clinical information of patients within the framework of machine learning for diabetes diagnostics. Therefore, time-consuming clinical testing using advanced medical equipment can be avoided, which is essential in communities with economic hardship or a lack of clinical expertise. In addition, the proposed study allows for elaborating on the most important information within patient profiles when testing for each microvascular complication. To the best of the authors’ knowledge, there have been very limited attempts towards identifying certain types of microvascular complications using machine learning. Therefore, a gap still exists in the literature about how certain patient information impacts the discrimination between diabetes complications. The present study provides a complete clinical testing approach for CAN-, DPN-, and RET-positive cases by looking into patient information from a machine-based perspective. NEP cases were not used in a separate machine-learning-based testing scenario because they can be easily identified from their patient profile information. Further, with a focus on CAN cases, the study investigates the ability of trained models to deeply discriminate between CAN-only patients and patients with additional complications alongside CAN.

## 2. Materials and Methods

### 2.1. Study Type

This is a cross-sectional study of Bangladeshi patients from Dhaka who have had type 2 diabetes mellitus for more than 10 years. We followed the STROBE cross-sectional reporting guidelines [36]. The study was approved by the ethical review committee of the Bangladesh University of Health Sciences (BUHS/BIO/EA/17/01) and conforms to the ethical principles outlined in the declaration of Helsinki and the Ministry of Health and Family Welfare of Bangladesh.

### 2.2. Inclusion and Exclusion Criteria

The parameters that were included in the inclusion criteria: Bangladeshi national, diagnosis of type 2 diabetes mellitus, above 40 years of age, able to give written consent, and the diabetes duration was 10 years or more. The exclusion criteria included: stroke history, having any heart disease, not being able to give consent, diabetes duration of less than 10 years, and the presence of any other pathophysiology that may lead to one or more similar complications, such as cancer.

### 2.3. Participants and Complications

One hundred and three (47 males and 56 females) unrelated patients of more than 40 years of age that had type 2 diabetes for 10 years or more were randomly selected and enrolled in the study during routine visits to the BIHS [37] Hospital between 18 December 2017 and 26 April 1018. This hospital is one of the most visited hospitals for diabetic patients in Bangladesh.

In this study, the recruited patients were diagnosed with complications, such as CAN, DPN, NEP, and RET (Table 1). The presence of these complications was confirmed by a qualified physician, based on the criteria outlined by the report of the WHO consultation group [38]. A diagnosis of cardiac autonomic neuropathy (CAN) was obtained from the Ewing test, which included five tests: deep breathing, lying to standing, the Valsalva maneuver, lying to standing BP, and sustained handgrip BP [39]. A diagnosis of diabetic peripheral neuropathy (NCV) was obtained using a nerve conduction velocity (NCV) test. There were several tests for recognizing polyneuropathy, CTS (carpal tunnel syndrome), peroneal neuropathy, and other types of neuropathies. A diagnosis of nephropathy (NEP) was determined by the ACR (albumin–creatinine ratio) level >30 mg/mmol for microalbuminuria, and >300 mg/mmol for macroalbuminuria [40]. A diagnosis of retinopathy (RET) was obtained from the fundus image test and classified according to the WHO criteria [41]. Fundus imaging is a process where 3-D retinal semi-transparent tissues are projected onto the imaging plane using reflected light and represented in 2-D [42].

Among these subjects, 70 were able to complete the diagnostic tests for all three complications (CAN, DPN, and RET). There were several combined complications found in some patients. The frequency of complications is shown in Table 2. To observe the importance of demographic, clinical, and laboratory profiles, a multiclass analysis (3-class analysis) was conducted using the classes marked in bold in Table 2 (CAN vs. CAN + DPN vs. CAN + DPN + Others). CAN + DPN + Others are the combinations of CAN + DPN + NEP, CAN + DPN + RET, and CAN + DPN + NEP + RET. These three classes were selected from Table 2 with higher numerals.

### 2.4. Types of Variables

#### 2.4.1. Demographic and Clinical Variables

The demographic data were collected from the patients at the time of enrollment. We measured the waist circumference, height, and weight at the time of enrollment and listed the value for the diabetic duration, age, gender, smoking history, and smokeless tobacco history. All of these data were verified from the necessary and relevant documents. The clinical data were measured at the time of enrollment. The blood pressure was measured on the first day before starting their Ewing test. If the systolic blood pressure was >130 mm Hg and diastolic blood pressure was >80 mm Hg or they were taking antihypertensive medications, it was called hypertension. Dyslipidemia was diagnosed from the medications of the patient or by checking the history of dyslipidemia of that patient. The data and its basic analysis are shown in Table 3.

#### 2.4.2. Laboratory Data

The laboratory data were taken from the laboratory of the hospital after the enrollment. The laboratory test parameters were hemoglobin A1c (HbA1c), microalbuminuria, urinary creatinine, and the albumin–creatinine ratio. The data and its basic analysis are shown in Table 4.

### 2.5. Machine Learning Modeling

#### 2.5.1. Clinical Testing Approach

To provide a complete diagnosis of a type 2 diabetes patient, four tests in two steps were applied sequentially (Figure 2) on patients’ demographic, clinical, and laboratory (DCL) information. This study supports type 2 diabetic patients with microvascular complications having a better screening from their DCL information. The approach combines a single-class binary classification model with three different classifiers and a multiclass classification model. The single-class classification model can run three tests in parallel to classify CAN, DPN, and RET separately. If all three tests result in a negative class, it means the patient with type 2 diabetes has no microvascular complications. If the test shows positive results, the patient goes for that specific complication treatment. However, obtaining a positive class from the CAN test leads to a multiclass classification model. This model can determine whether the patient has other microvascular complications along with CAN. Thus, this results of this model include: CAN (having only CAN), CANDPN (having DPN with CAN), CANDPN+ (having NEP or RET with CAN and DPN). The resulting class determines the treatment that should be provided to the patient.

#### 2.5.2. Analysis of the Demographic Clinical and Laboratory Profiles

The demographic variables (such as gender, height, age, weight, smoking history, tobacco history, and diabetes duration), clinical measurements (waist circumference, BMI, systolic blood pressure, and diastolic blood pressure), and measured laboratory values (such as HbA1c, microalbuminuria, urinary creatinine, and albumin–creatinine ratio) were selected for further analysis as patient information.

A feature selection approach was then followed based on the univariate chi-squared test to choose the foremost critical factors among all the demographic, clinical, and laboratory variables. In this test, a statistical hypothesis investigation is performed for each DCL feature to test whether the observed calculations coordinate with the anticipated ones, i.e., patient’s complication type. Moreover, it gives a noteworthy distinction *p*-value measure (*p*-value < 0.05) between categories based on the statistical calculations and desire [43]. A feature with a lower *p*-value signifies that this variable is most likely dependent on the complication label. Hence, it is vital for anticipating the complication and has discriminatory characteristics. In this way, a score of significance is returned for each DCL profile utilized within the test as score = −log (*p*). In this work, we call this score importance. We calculated importance using a function called fscchi2 () in MATLAB 2021a.

#### 2.5.3. Support Vector Machine (SVM)

SVM is an exceedingly popular machine learning algorithm used in classification and regression problems. It is one of the classic machine learning techniques that can help to solve big data classification problems. SVM allows the classification of single-class as well as multiclass classification problems. It is commonly utilized as an exception finder, where the model is prepared to recognize training data from any other irrelevant information [44]. The model tends to distinguish which unused objects are closely representing the selected class in the training phase, which is generally called a positive class [45]. A set of probabilities has been returned by the model to show the degree of matching between the testing and training samples. In this paper, a single-class SVM was used for the training model in the CAN, DPN, and RET tests. Having the complication has been considered as the positive class in the single-class classification. However, a multiclass SVM was for training in the CANDPNOthers test. To guarantee the highest performance from the model, a non-linear RBF (radial basis function) kernel was used with fine-tuned hyper-parameters.

#### 2.5.4. Random Forest (RF)

Random forest (RF), also known as classification and regression tree (CART), is a form of decision trees, where a set of tree-like trait nodes is associated with a set of sub-trees of decision nodes [46,47]. This algorithm is considered a conglomeration strategy that employs the concepts of bagging. All the decision trees are calculated based on the corresponding resource cost, outcome chances, and utility to provide a prediction. The prediction preparation begins by doling out an occasion at each tree to its root node. At that point, for each of the subsequent sub-nodes, the results are calculated successively. Once a leaf is experienced, the tree-like nodes halt and an occasion is relegated with a prediction. All of the occasions and predictions shape the ultimate choice made by the tree model [48]. In this work, 20–120 decision trees were utilized to construct the model. The choice of the number of trees for each single-class test, as well as the multiclass test, was fine-tuned to guarantee the greatest conceivable performance from the model.

#### 2.5.5. Logistic Regression

Logistic regression is one of the most commonly used machine learning algorithms in statistics. It is a statistical model that uses a logistic function to represent a binary dependent variable in its most basic form, though there are many more advanced variants. Logistic regression is a technique for estimating the parameters of a logistic model in regression analysis. The natural logarithm of the odds is used as a regression function of the predictors in the logistic regression model. The expression for a one predictor (X) one outcome (Y) logistic regression model is ln [odds (Y = 1)] = β0 + β1X, where ln is the natural algorithm, Y = 1 or Y = 0 refers to the event occurrence of the event, β0 is the intercept term, and β1 is the regression coefficient that refers to the change in the logarithm of the event’s odds with a 1-unit change in the predictor X [49].

#### 2.5.6. Training and Testing

A leave-one-out scheme was followed in the single-class models, as well as in the multiclass model, to ensure the incorporation of the highest possible number of samples within the prepared models. Besides, it was fundamental to supply a prediction for each and every patient. An iterative process was applied in this scheme by selecting one subject as testing data, whereas the remaining subjects were used for training. The method repeated on each cycle until a prediction was given for every subject.

#### 2.5.7. Parameter Optimization

In each test, several model parameters were fine-tuned to ensure the highest acquirable model performance. Performance was measured in the form of accuracy, sensitivity, specificity, precision, f1-score, and area under the curve (AUC). To handle data imbalance (65 positive classes vs. 10 negative classes in the CAN test and 7 positive classes vs. 89 negative classes in the RET test), a model parameter called ‘prior probability’ was introduced in the algorithm during the training phase. The prior probabilities were found observationally, where the initial weight was set to each class that was equal to its number of samples relative to the whole number of samples [50]. Prior probability was not used in the DPN test, as it had balanced classes. The minimum leaf size and bag fraction value were used as per the behavior of the RF model, on an iterative basis and keeping the optimum value.

## 3. Results

### 3.1. Demographic, Clinical, and Laboratory Profiles

Demographic and clinical data, along with major comorbidities with type 2 diabetes, are shown in Table 3, and laboratory profiles are shown in Table 4. There were 47 (45.63%) male patients and 56 (54.37%) female patients. The mean age of the patients was 56 years (±8.913), the mean ages of the male and female patients were 57.1 years (±9.78) and 54.6 years (±7.93), respectively. This is consistent with the finding that the diabetic population in Bangladesh, as well as south Asia, are comparatively younger than in the west [51,52]. The sub-variables under ‘Age’ show that 46.8% of the male subjects were greater than 60 years old, but about 40% female subjects were between 40 and 50, though, overall, the patients showed an increasing prevalence for a higher age. A study in Spain also showed that an increase in patient age increases the prevalence of diabetic complications [19]. In this study, 27 (57.45%) males, 35 (62.50%) females, and a total of 62 (60.19%) patients had a history of hypertension (mean systolic blood pressure was 138.4 mm Hg). A total of 35 (33.98%) patients had dyslipidemia, where 14 (29.79%) were male and 21 (37.5%) were female. Only nine (8.74%) patients had a history of smoking, and they were all male. In addition, the overweight condition (42.86%) was common for female diabetic patients, with more than 98% female subjects having a waist circumference higher than 80 cm, while 57.45% of the male subjects had a normal weight. Though obesity was relatively common for female patients (27%), a total of 20 (19.42%) patients were obese (mean BMI (body mass index) = 33.94 kg/m^2^ and mean waist circumference = 90.84 cm for males and 97.38 cm for females). For the retinopathy patients, the waist circumference was 89.91 cm for males and 93.98 cm for females, where 15 (26.79%) were female and 5 (10.638%) were males.

More than 67% of the patients for any type of complication had a high HbA1c (mean HbA1c = 8.824, male mean HbA1c = 9.066, and female mean HbA1c = 8.621 for the patients with CAN). The retinopathy patients had very high HbA1c (mean HbA1c = 10.829, male mean HbA1c = 10.720, and female mean HbA1c = 11.100). Microalbuminuria was found in 25 (24.51%) patients, where 10 were male and 15 were female. In the case of nephropathy, a total of 22 (78.57%) patients had microalbuminuria. All the retinopathy patients had a creatinine level of 20 to 320 mg/dL. The mean ACR (albumin–creatinine ratio) for the patients was 35.967 mg/mmol, where 47 (46.08%) males had a mean ACR of 32.092 mg/mmol, and 55 (53.92%) females had a mean ACR of 39.280 mg/mmol. Neuropathy was the most common complication in Bangladeshi diabetic type 2 patients of more than 40 years’ old who had diabetes for more than 10 years. Besides, there were very few retinopathy patients, so it implies that the rate of retinopathy in Bangladeshi type 2 diabetes patients is very low.

### 3.2. Complications of Type 2 Diabetes

Overall, more than one clinically diagnosed complication was present in 99 subjects out of the cohort of 103 diabetics included in this study. Most of the subjects had CAN (66.02%), followed by diabetic peripheral neuropathy (43.69%), nephropathy (27.18%), and retinopathy (6.8%). Those patients who had retinopathy also had CAN and DPN. The rate of retinopathy complication was very low. Only seven retinopathy patients were found, and five patients out of them had all types of complication, while the other two had CAN and DPN. This trend suggests that RET should be the final stage of the above four diabetes microvascular complications in Bangladesh. We did not find any subject with only NEP or only RET. If a patient had RET, we can say that he/she had CAN and DPN both or CAN, DPN, and NEP, i.e., all the complications. The average diabetic duration of the male patients with CAN and DPN was high (17.33 years for CAN and 18.91 years for DPN) and comparatively lower for RET (13 years). The female patients with retinopathy had a high diabetic duration of 17.5 years. They did not check for DM until they became very ill, so their reported DM duration is from the day they first found out, not from the actual moment of DM development. The overall result indicates a high prevalence of complications in Bangladeshi type 2 diabetes patients.

### 3.3. Classification of Cardiac-Related Microvascular Complications

To assess the association between any complication (as an outcome) and significant demographic, clinical, and laboratory variables of the patients, several machine learning models (logistic regression, RF, and SVM) were trained by changing the model parameters in an iterative way and observing the sensitivity, specificity, precision, f1-score, and accuracy of the model. The chi-squared (χ^2^) test was used to choose significant variables and we use only these significant variables to determine the classification accuracy. The threshold for a significant importance level was different for each test.

#### 3.3.1. CAN

We found diastolic BP (importance 2.1), albumin–creatinine ratio (importance 1.6), and gender (importance 1) to be the significant predictors for screening CAN, which is the most common complication among Bangladeshi patients with type 2 diabetes. We had 65 positive and 10 negative CAN patients in our study. To find the best suitable result and to maximize the model performance, we used prior probability in the classification model of CAN. We found that RF was the best model at the weight of (1.05 and 0.9). The performance (shown in Table 5) of the model was obtained as accuracy 98.68%, sensitivity 98.48%, and specificity 100%. The performance is shown in Figure 3. In Table 6, the 95% confidence intervals, including the mean values of the features, are provided for CAN patients to represent the true mean of the population.

#### 3.3.2. DPN

Similarly, microalbuminuria (importance 5.1), smoking history (importance 2.9), smokeless tobacco history (importance 2.7), HbA1c (importance 2.4), albumin–creatinine ratio (importance 1.9), systolic BP (importance 1.8), diastolic BP (importance 1.4), and urinary creatinine (importance 1.4) were found to be the most significant predictors for determining DPN from type 2 diabetes patients in Bangladesh. This is consistent with other findings that age and diabetic duration are insignificant [15,53,54,55,56,57,58] here, since all the patients were more than 40 years of age and the diabetic duration was a minimum of 10 years. Both the RF and SVM models showed the highest accuracy for classifying DPN in the patients with type 2 diabetes mellitus from Bangladesh. Figure 4 illustrates the result of classifying DPN, and the numeric values are stored in Table 5. Table 7 shows the true means and 95% confidence intervals of the populations included in the DPN test.

#### 3.3.3. RET

In the case of diabetic retinopathy (RET), HbA1c (importance 6.1), microalbuminuria (importance 4.7), smokeless tobacco history (importance 2.8), weight (importance 1.9), gender (importance 1.8), urinary creatinine (importance 1.7), and albumin–creatinine ratio (importance 1.7) were found to be significant predictors to classify whether a type 2 diabetes mellitus patient has retinopathy (Figure 5). A previous study in Bangladesh showed a 5.4% prevalence of retinopathy patients [22], and in our study, we had 6.8% of type 2 diabetes patients with retinopathy. The accuracy (shown in Table 5) of the RF model was 84.38%. To show the true mean of the features in diabetic retinopathy test, Table 8 is added with means and 95% confidence interval information.

#### 3.3.4. CANDPNOthers

The multiclass analysis test provides a comprehensive picture of patients who have other diabetic neuropathies in addition to CAN. A total of 55 patients were considered as training and testing inputs for machine learning models, with 16 suffering from CANDPN and 14 suffering from CANDPN+, where ‘Others’ included NEP and RET. In the CANDPNOther test, three classes were assigned to the model, i.e., CAN vs. CANDPN (the patients with both CAN and DPN complications) vs. CANDPN+ (the patients with CAN, DPN, and NEP; CAN, DPN, and RET; or CAN, DPN, NEP, and RET). We only included these classes due to the insufficient number of patients in the other classes. SVM performed better in this multiclass classification rather than RF. The confusion matrix (Figure 6) illustrates that the CAN and CANDPN+ classes could be classified effectively. However, identifying CANDPN patients using this model might be inefficient. The features used in this model were the albumin–creatinine ratio and microalbuminuria. These two features are common for all the binary tests that have been performed in this study.

## 4. Discussion

This study demonstrated the importance of demographic, clinical, and laboratory profiles in the machine learning domain for the classification of diabetic microvascular complications. Moreover, this study illustrated a complete machine-learning-based clinical approach to screen diabetic patients suffering from diabetic microvascular complications. It also provides an association of other microvascular complications along with CAN. It provides a stepwise clinical approach to screen diabetes microvascular complications for the Bangladeshi type 2 diabetic cohort. The high performance achieved in each test strongly suggests that DCL profiles should be included as features in the machine learning approach to ensure a high classification accuracy. In our study, we also showed the DCL profiles that are highly associated with a kind of complication. Thus, the clinician can easily coordinate the physiological grounds.

### 4.1. Demographic, Clinical, and Laboratory Profiles

In this study, we demonstrated the significance of DCL profiles to screen a microvascular complication of the type 2 diabetic population in Bangladesh. The profiles that were used in this study are easily collectible by any hospital in Bangladesh. Moreover, gathering this information from a patient is not costly. Furthermore, all the DCL profiles used in this study are not required to be collected for screening using the proposed method. Only the significant features that are listed (Section 3.3) for each test will be needed to execute the test. However, to execute all the tests proposed in this study, a mathematical union of all the significant features that are used in each test would be required.

Diastolic BP, Albumin–Creatinine Ratio, and Gender were highly associated with CAN in our CAN test. Thus, we have found the highest accuracy for the CAN classifier by using these three predictors. In [59,60], the authors showed the influence of hypertension on diabetic complications, and our study we found that diastolic BP was a good predictor variable to classify CAN. However, we have found that HbA1c was not required for testing CAN. On the other hand, microalbuminuria was significantly associated with peripheral neuropathy, nephropathy, and retinopathy. This finding supports both the studies in [61,62], where the authors showed the association of microalbuminuria with nephropathy and retinopathy. Moreover, in [63], Bell et. al. observed the significant association of microalbuminuria with diabetic neuropathy. HbA1c was significantly associated with retinopathy in our study. It was also associated with peripheral neuropathy. In [64,65], the authors established the relationship between HbA1c and microvascular complications. Many authors showed the significance and association of different demographic, clinical, and laboratory parameters with diabetic microvascular complications. However, in our study, we found significant DCL profiles using a statistical model and used these significant profiles with a machine learning model to show the performance.

### 4.2. Machine Learning as a Screening Tool

This work describes the application of a modern machine learning model, combining the use of statistically significant features to exploit demographic, clinical, and laboratory data to extract a classifier that can classify type 2 diabetes microvascular complications. To address the class imbalance, a machine learning hyper-parameter ‘prior probability’ was used. Picking up the benefits of the recent advances of machine learning in the area of diabetes diagnosis is considered to be fundamental. It makes a difference within the investigation of colossal healthcare records and changes them into clinical experiences that can help healthcare experts in prompt and intelligent decision-making. Even though the involvement of a clinician within the diagnosis and treatment of diabetic patients may be necessary, machine learning models might be able to provide an early-stage screening that can avoid numerous complications from further development. Besides, when there is a tremendous request for medical specialists or unbounded data available, it is quite hard to provide a complete diagnostic for each quite effectively. In this manner, pre-trained machine learning models can make the process faster and less rigorous for healthcare suppliers and practitioners. Different sorts of machine learning algorithms, such as support vector machines (SVM), K-nearest neighbor (KNN), choice trees, etc., have been utilized broadly within the research associated with type 2 diabetes microvascular complications [66].

CAN plays a major role in myocardial ischemia and infarction, heart arrhythmias, hypertension, and heart disappointment and it increases the risk of sudden cardiac death. Jelinek and Cornforth [67] proposed a novel clustering technique using a graph-based machine learning system that enables the identification of severe diabetic neuropathies in 2016. This proposed model outperforms SVM, RF, and KNN. Cho et al. [68] showed an accuracy of 88.7% (AUC 0.969 and specificity 0.85) using SVM classifiers along with a feature selection method for the prediction of diabetic nephropathy from the data of 4321 patients. Reedy et. al. [69] proposed a multi-model ensemble-based machine learning algorithm to classify diabetic retinopathy. The authors included several machine learning classifiers in their research and concluded that the ensemble model provides better accuracy with better sensitivity and specificity. Sambyal et. al. [66] provided a review of using machine learning models to classify diabetes microvascular complications in 2020. The authors showed that most of the work for classifying RET had been conducted using fundus image as the input, and he compared the different achieved accuracies of the different classifiers by different authors. By only using the demographic, clinical, and laboratory profiles, our model outperforms all the models reviewed by the author in terms of classifying diabetic retinopathy. The authors also have reviewed several machine learning models proposed by different authors for classifying cardiac autonomic neuropathy and nephropathy. However, we only used DCL profiles as independent features to classify microvascular complications.

### 4.3. Clinical Relevance

The test schemes followed in this work offer physicians an important clinical diagnostic method in the evaluation and diagnosis of type 2 diabetes microvascular complications. The single-class classifiers can operate individually in parallel or sequentially. However, for finding combined complications with CAN, the model works sequentially with the CAN testing classifier. Since this model is sequential, this digs deeper by analyzing diabetics with single and combined complications. Single-class classifiers identify any microvascular complication that is present in a patient, regardless of whether the multiclass classifier predicts the presence of other complications with CAN. Such a clinical test will ensure a better diagnose of type 2 diabetes microvascular complications by distinguishing the cause of single CAN and other related complications. Furthermore, the silent nature of these complications makes it difficult to diagnosis correctly, especially when combined with other microvascular complications. The performance achieved through machine learning using only DCL profiles in this study provides a path to prevent many undiagnosed CAN-only cases. Since a CAN-only medical procedure may not provide effective treatment if additional complications are not properly identified. It is vital to know about combined complications. The multiclass classification test helps to identify multiple complications with autonomic neuropathy.

### 4.4. Key Message to the Health Community of Bangladesh

Globally, healthcare stakeholders are entering a new era of data-driven clinical detection and prognostication. The application of modern machine-learning-based approaches offers great promises for early diagnosis or prognosis of various health complications. The early identification of patients at risk of microvascular complications due to type 2 diabetes can mitigate the burden on the healthcare system, especially in the context of a resource-limited setup. As the present study shows that screening is feasible from the demographic, clinical, and laboratory (DCL) variables using a proper machine learning classification model, the health community can utilize this benefit for screening that can avoid numerous complications from further development. It also can help healthcare experts in prompt and intelligent decision-making and save the patients from incurring greater healthcare costs.

## 5. Conclusions

This study explored the present status of microvascular complications in a cohort of type 2 diabetes patients in Bangladesh. Higher comorbidities and microvascular complications were found as compared with neighboring countries, most likely due to the increased levels of hypertension in this cohort. This study also suggests that a high diastolic BP and albumin-creatinine ratio are related to CAN; high microalbuminuria, HbA1c, and blood pressure are related to DPN; high HbA1c and microalbuminuria are related to RET. These findings may be useful in finding risk factors for the development of diabetic complications. Using these risk factors as the independent features, a machine learning model could be designed to screen microvascular complications. This study shows a machine learning model could be utilized to identify diabetes complications in Bangladesh, where the majority of its population is poor. We believe this study could contribute to more effective and affordable screening techniques [70] for diabetes-related microvascular complications.

It is worth noting that the proposed study should be further validated on a wider patient cohort to strengthen the observations. Although the findings of this study were promising and correlate with the observations found in the literature, one limitation to the current work was the small sample size, which is a common situation in biomedical studies that rely on patient data. Overall, RF and SVM are known to handle small sample sizes with high performance capabilities [71,72,73], especially when compared to other artificial intelligence algorithms, such as deep neural networks, that require large datasets. Therefore, an essential future direction to the current study is to be tested on large clinical data and with additional machine/deep learning algorithms.

## Figures and Tables

**Figure 1 jcm-11-00903-f001:**
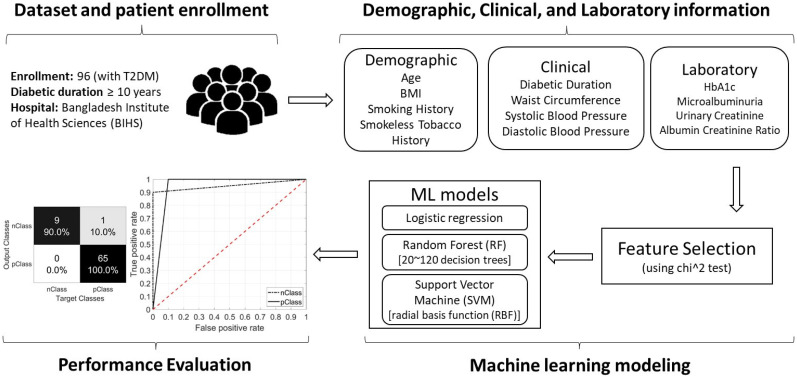
A graphical view of the complete research work in this study, including patient enrollment; demographic, clinical, and laboratory information acquisition; machine learning modeling; and performance evaluation of the model.

**Figure 2 jcm-11-00903-f002:**
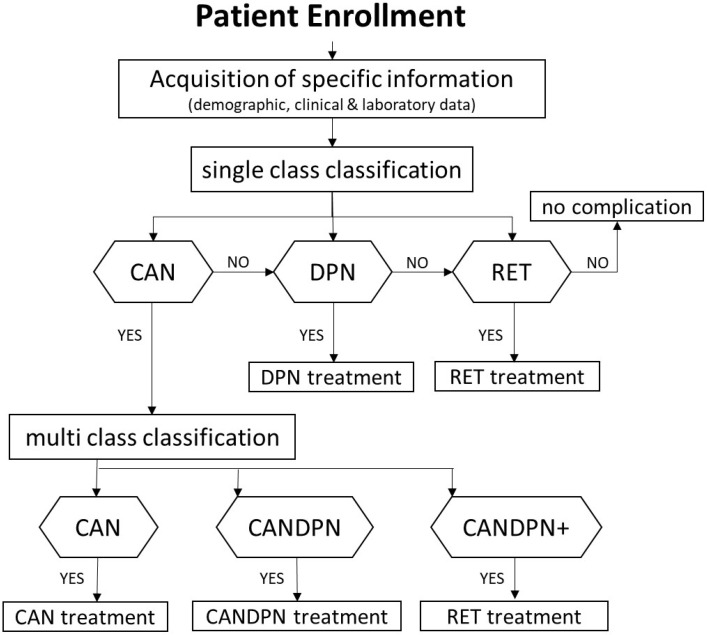
The proposed procedure for screening diabetic patients. Every patient initially goes through the information acquisition of this clinical diagnosis flowchart. Five tests are then applied in two stages. The second stage (multiclass class is only for the patients who go through the CAN test and have a positive CAN. A single-class classification can predict the presence of microvascular complications (CAN, DPN, or RET) and can predict whether there is any presence of complications. Multiple complications with CAN could be classified using the multiclass classifier.

**Figure 3 jcm-11-00903-f003:**
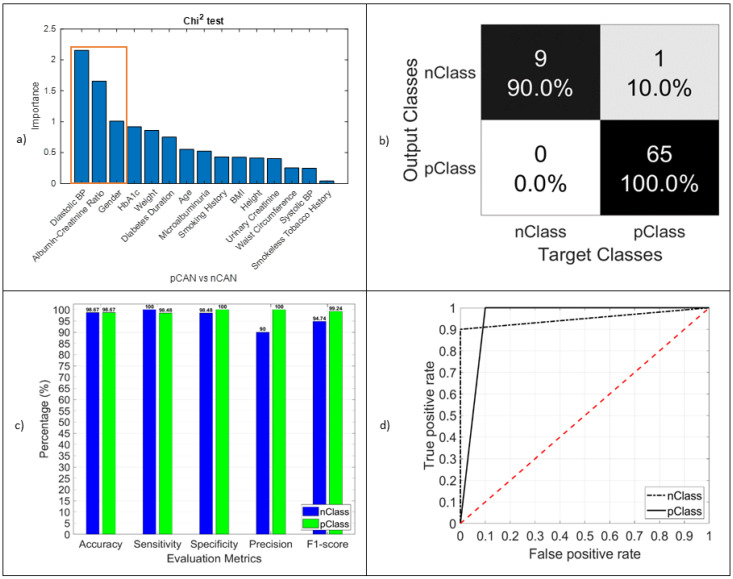
(**a**) Chi-squared test result. The importance of different marked features was used in the model as an identifier; (**b**) confusion matrix of the CAN test (pClass vs. nClass); (**c**) performance evaluation matrices; (**d**) TPR vs. FPR, graphical view of the CAN classifier model performance.

**Figure 4 jcm-11-00903-f004:**
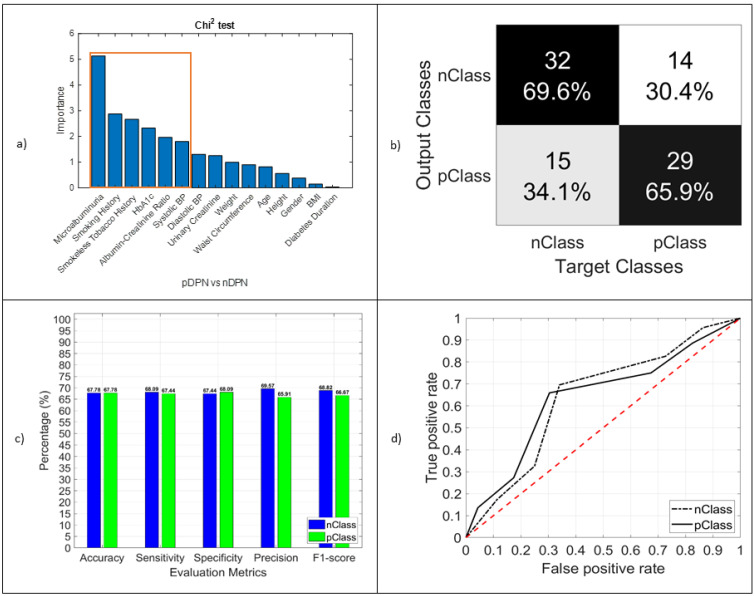
(**a**) Chi-squared test result. The importance of different marked features was used in the model as an identifier; (**b**) confusion matrix of the DPN test (pClass vs. nClass); (**c**) performance evaluation matrices; (**d**) TPR vs. FPR, graphical view of the DPN classifier model performance.

**Figure 5 jcm-11-00903-f005:**
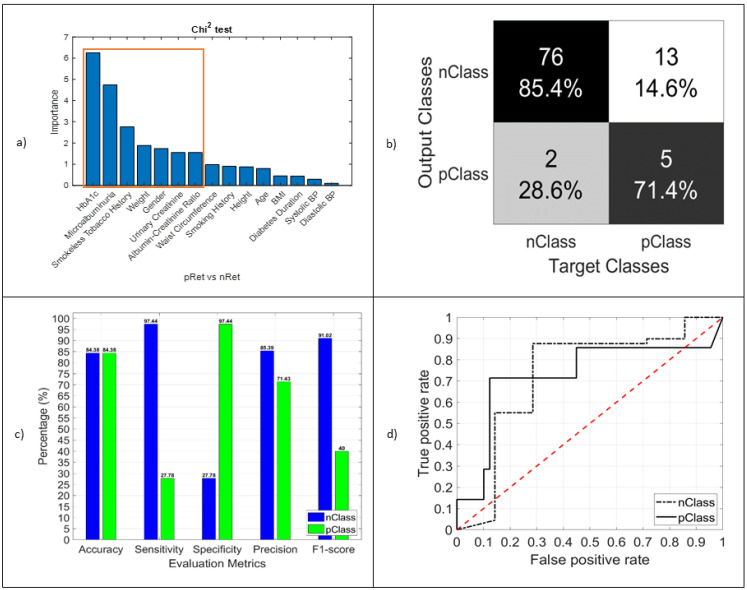
(**a**) Chi-squared test result. The importance of different marked features was used in the model as an identifier; (**b**) confusion matrix of the RET test (pClass vs. nClass); (**c**) performance evaluation matrices; (**d**) TPR vs. FPR, graphical view of the RET classifier model performance.

**Figure 6 jcm-11-00903-f006:**
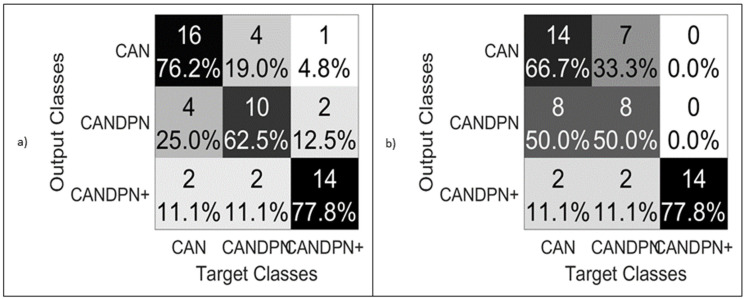
Performance comparison (confusion matrix) between the multiclass SVM classifier and the multiclass RF classifier. (**a**) Confusion matrix of SVM classifier; (**b**) confusion matrix of RF classifier; classes: 1. CAN, patients with CAN; 2. CANDPN, with DPN and CAN; 3. CANDPN+, patients with NEP and/or RET with CAN and DPN.

**Table 1 jcm-11-00903-t001:** Types of complications of patients included in this study.

Name of the Complication	Type	Number of Patients, *N* (%)	Total, *N*
CAN	pCAN (with CAN)	65 (67.708)	96
	nCAN (without CAN)	10 (10.417)	
	Test result unavailable	21 (21.875)	
DPN	pDPN (with DPN)	44 (45.833)	96
	nDPN (without DPN)	46 (47.917)	
	Test result unavailable	6 (6.250)	
RET	pRET (with RET)	7 (7.292)	96
	nRET (without RET)	89 (92.708)	

**Table 2 jcm-11-00903-t002:** Types and frequency of complications of diabetes patients.

Types of Complications		Numerals, *N* (%)	Total, *N*
**nComp (no complication)**		4 (4.16)	96
**Single Complications**	**CAN**	**21 (21.875)**	
	DPN	3 (3.125)	
	NEP	0 (0.00)	
	RET	0 (0.00)	
**Combined Complications**	**CAN and DPN**	**16 (16.67)**	
	CAN and NEP	6 (6.25)	
	DPN and NEP	2 (2.083)	
	**CAN, DPN, and NEP**	**12 (12.5)**	
	**CAN, DPN, and RET**	**2 (2.083)**	
	**CAN, DPN, NEP, and RET**	**4 (4.16)**	
**Not sure (due to unavailable test results)**		26 (27.08)	

**Table 3 jcm-11-00903-t003:** Demographic and clinical variables of patients.

Demographic Variables						
** *Variables and* ** ** *their subdivisions* **	Male		Female		All	
	Mean ± SD	N (%of M)	Mean ± SD	N (% of F)	Mean ± SD	N (% of total)
*Patients*		47 (45.63)		56 (54.37)		103 (100)
*Age (years)*	57.70 ± 9.78	47 (100)	54.60 ± 7.93	56 (100)	56.01 ± 8.91	103 (100)
≥40 and <50	44.8 ± 3.22	*10 (21.28)*	45.6 ± 2.95	*15 (26.79)*	45.28 ± 3.02	*25 (24.27)*
≥50 and <60	53.2 ± 2.7	*15 (31.91)*	52.86 ± 3.17	*22 (39.29)*	53 ± 2.95	*37 (35.92)*
≥60	66.63 ± 4.78	*22 (46.81)*	63.73 ± 3.79	*19 (33.93)*	65.29 ± 4.54	*41 (39.80)*
CAN	58.74 ± 9.63	31 (65.95)	53.32 ± 7.40	37 (66.07)	55.79 ± 8.85	68 (66.01)
DPN	58.95 ± 10.33	21 (44.68)	52.58 ± 6.33	24 (42.85)	55.55 ± 8.93	45 (43.68)
Nep	58.5 ± 10.37	12 (25.53)	54.37 ± 8.75	16 (28.57)	56.14 ± 9.52	28 (27.18)
Ret	56.8 ± 11.64	5 (10.63)	47.5 ± 0.707	2 (3.571)	54.14 ± 10.54	7 (6.796)
*BMI (kg/m^2^)*	25.53 ± 3.47	47 (100)	27.93 ± 5.08	56 (100)	26.84 ± 4.56	103 (100)
Underweight: <18.5	0	0 (0)	0	0 (0)	0	0 (0)
Normal: ≥18.5, <25	23.54 ± 1.45	**27 (57.45)**	22.93 ± 1.69	17 (30.36)	23.31 ± 1.56	44 (42.72)
Overweight: ≥25.0, <30	26.54 ± 1.03	15 (31.91)	27.58 ± 1.32	**24 (42.86)**	27.18 ± 1.31	39 (37.86)
Obese: ≥30	33.23 ± 4.09	5 (10.638)	34.18 ± 4.77	15 (26.79)	33.94 ± 4.52	20 (19.42)
CAN	26.26 ± 3.71	31 (65.95)	27.94 ± 5.82	37 (66.07)	27.17 ± 5.01	68 (66.01)
DPN	25.52 ± 3.56	21 (44.68)	28.75 ± 5.01	24 (42.85)	27.24 ± 4.64	45 (43.68)
Nep	26.17 ± 4.22	12 (25.53)	29.18 ± 5.60	16 (28.57)	27.89 ± 5.19	28 (27.18)
Ret	26.79 ± 5.53	5 (10.63)	26.29 ± 2.09	2 (3.571)	26.65 ± 4.60	7 (6.796)
*Smoking history*		9 (19.15)		0 (0)		**9 (8.74)**
*Smokeless tobacco history*		10 (21.28)		17 (30.357)		**27 (26.21)**
**Clinical variables**						
** *Name of the Variables and* ** ** *their subdivisions* **	Male		Female		All	
	Mean ± SD	N (%of M)	Mean ± SD	N (% of F)	Mean ± SD	N (% of total)
*Diabetes duration (years)*	16.17 ± 6.07	47 (100)	15.55 ± 5.76	56 (100)	15.83 ± 5.88	103 (100)
≥10 and <20	13.54 ± 2.76	37 (78.72)	12.60 ± 2.64	41 (73.21)	13.05 ± 2.73	78 (75.73)
≥20 and <30	24 ± 3.116	8 (17.02)	22.30 ± 1.93	13 (23.21)	22.95 ± 2.52	21 (20.39)
≥30	33.5 ± 2.12	2 (4.26)	32 ± 2.828	2 (3.57)	32.75 ± 2.22	4 (3.88)
CAN	16.54 ± 6.20	31 (65.95)	16.13 ± 6.01	37 (66.07)	16.32 ± 6.05	68 (66.01)
DPN	17.33 ± 7.43	21 (44.68)	**14.16 ± 4.80**	24 (42.85)	15.64 ± 6.30	45 (43.68)
Nep	18.91 ± 8.11	12 (25.53)	16.81 ± 6.63	16 (28.57)	17.71 ± 7.24	28 (27.18)
Ret	13 ± 2.828	5 (10.63)	17.5 ± 3.535	2 (3.571)	14.28 ± 3.49	7 (6.796)
*Waist Circumference (cm)*	90.84 ± 8.61	47 (100)	97.38 ± 9.46	56 (100)	94.39 ± 9.61	103 (100)
Men ≥90	97.40 ± 6.7	23 (48.94)				
Women ≥80			97.72 ± 9.19	**55 (98.21)**		
CAN	92.09 ± 8.47	31 (65.95)	96.58 ± 9.30	37 (66.07)	94.54 ± 9.15	68 (66.01)
DPN	92.64 ± 8.13	21 (44.68)	**98.63 ± 9.07**	24 (42.85)	95.84 ± 9.06	45 (43.68)
Nep	91.22 ± 6.71	12 (25.53)	97.31 ± 9.80	16 (28.57)	94.70 ± 9.00	28 (27.18)
Ret	89.91 ± 5.26	5 (10.63)	93.98 ± 14.36	2 (3.571)	91.07 ± 7.53	7 (6.796)
*Systolic blood pressure (mmHg)*	141.2 ± 19.5	47 (100)	136.0 ± 20.14	56 (100)	138.4 ± 19.94	103 (100)
≤119	108 ± 5.29	4 (8.51)	108.3 ± 8.96	12 (21.43)	108.2 ± 8.03	16 (15.53)
≥120 and <14	129.2 ± 6.67	19 (40.43)	130.1 ± 4.98	19 (33.93)	129.7 ± 5.82	38 (36.89)
≥140 and <160	148.2 ± 7.52	15 (31.91)	148.3 ± 5.71	19 (33.93)	148.2 ± 6.47	34 (33.01)
≥160	169.6 ± 9.72	9 (19.15)	171.3 ± 6.40	6 (10.714)	170.3 ± 8.33	15 (14.56)
CAN	145.0 ± 20.16	31 (65.95)	134.0 ± 21.30	37 (66.07)	139.0 ± 21.35	68 (66.01)
DPN	148.5 ± 20.82	21 (44.68)	134.8 ± 15.96	24 (42.85)	141.2 ± 19.43	45 (43.68)
Nep	153.0 ± 15.16	12 (25.53)	136.1 ± 17.22	16 (28.57)	143.4 ± 18.19	28 (27.18)
Ret	158.6 ± 16.14	5 (10.63)	137.5 ± 17.67	2 (3.571)	152.5 ± 18.21	7 (6.796)
*Diastolic blood pressure (mmHg)*	78.97 ± 9.86	47 (100)	76.42 ± 11.96	56 (100)	77.59 ± 11.07	103 (100)
≤79	71.36 ± 7.45	22 (46.81)	67.96 ± 6.98	32 (57.14)	69.35 ± 7.30	54 (52.43)
≥80–89	82.73 ± 2.83	19 (40.43)	83.81 ± 3.08	16 (28.57)	83.22 ± 2.95	35 (33.98)
≥90–99	94 ± 3.39	5 (10.64)	94.14 ± 2.61	7 (12.5)	94.08 ± 2.81	12 (11.65)
≥100	100 ± 0	1 (2.13)	105 ± 0	1 (1.79)	102.5 ± 3.54	2 (1.94)
CAN	78.45 ± 11.40	31 (65.95)	75.48 ± 12.76	37 (66.07)	76.83 ± 12.16	68 (66.01)
DPN	78.19 ± 12.23	21 (44.68)	76.87 ± 10.63	24 (42.85)	77.48 ± 11.29	45 (43.68)
Nep	74.91 ± 13.48	12 (25.53)	75.93 ± 10.81	16 (28.57)	75.5 ± 11.79	28 (27.18)
Ret	84.6 ± 10.13	5 (10.63)	72.5 ± 3.54	2 (3.571)	81.14 ± 10.27	7 (6.796)

**Table 4 jcm-11-00903-t004:** Laboratory variables of patients.

Types and Their Variables	Male		Female		All	
	Mean ± SD	N (%of M)	Mean ± SD	N (% of F)	Mean ± SD	N (% of total)
**HbA1c (mmol/mol,%)**						
*Not specified*	9.066 ± 1.944	47 (45.63)	8.621 ± 1.453	56 (54.37)	8.824 ± 1.701	103 (100.0)
Optimal: <7		2 (4.26)		8 (14.29)		10 (9.71)
Fair: 7–8		12 (25.53)		11 (19.64)		23 (22.33)
High: >8		33 (70.21)		37 (66.07)		70 (67.96)
*CAN*	9.213 ± 1.790	31 (45.59)	8.716 ± 1.491	37 (54.41)	8.943 ± 1.640	68 (66.02)
Optimal: <7		1 (3.23)		4 (10.81)		5 (7.35)
Fair: 7–8		6 (19.35)		8 (21.62)		14 (20.59)
High: >8		24 (77.42)		25 (67.57)		49 (72.06)
*DPN*	9.291 ± 1.988	21 (46.67)	8.930 ± 1.667	24 (53.33)	9.098 ± 1.810	45 (43.69)
Optimal: <7		2 (9.52)		3 (12.50)		5 (11.11)
Fair: 7–8		3 (14.29)		4 (16.67)		7 (15.56)
High: >8		16 (76.19)		17 (70.83)		33 (73.33)
*Nephropathy*	9.9750 ± 2.221	12 (42.86)	8.763 ± 1.902	16 (57.14)	9.282 ± 2.094	28 (27.18)
Optimal: <7		1 (8.33)		3 (18.75)		4 (14.29)
Fair: 7–8		1 (8.33)		4 (25.00)		5 (17.86)
High: >8		10 (83.33)		9 (56.25)		19 (67.86)
*Retinopathy*	10.720 ± 3.334	5 (71.43)	11.100 ± 1.980	2 (28.57)	10.829 ± 2.846	7 (6.80)
Optimal: <7		0 (0.00)		0 (0.00)		0 (0.00)
Fair: 7–8		2 (40.00)		0 (0.00)		2 (28.57)
High: >8		3 (60.00)		2 (100.00)		5 (71.43)
**Microalbuminuria (mg)**						
*Not specified*	60.6164 ± 99.490	47 (46.08)	49.571 ± 82.123	55 (53.92)	54.661 ± 90.247	102 (99.03)
Optimal: <30		34 (72.34)		38 (69.09)		72 (70.59)
Microalbuminuria: 30–300		10 (21.28)		15 (27.27)		25 (24.51)
Macro albuminuria: >300		3 (6.38)		2 (3.64)		5 (4.90)
*CAN*	88.439 ± 113.172	31 (45.59)	56.981 ± 93.199	37 (54.41)	71.322 ± 103.204	68 (66.02)
Optimal: <30		18 (58.06)		25 (67.57)		43 (63.24)
Microalbuminuria: 30–300		10 (32.26)		10 (27.03)		20 (29.41)
Macro albuminuria: >300		3 (9.68)		2 (5.41)		5 (7.35)
*DPN*	121.925 ± 124.49	21 (47.73)	55.2565 ± 87.479	23 (52.27)	87.075 ± 110.720	44 (42.72)
Optimal: <30		10 (47.62)		15 (65.22)		25 (56.82)
Microalbuminuria: 30–300		8 (38.10)		7 (30.43)		15 (34.09)
Macro albuminuria: >300		3 (14.29)		1 (4.35)		4 (9.09)
*Nephropathy*	210.308 ± 91.414	12 (42.86)	144.519 ± 98.407	16 (57.14)	172.7143 ± 99.417	28 (27.18)
Optimal: <30		0 (0.00)		1 (6.25)		1 (3.57)
Microalbuminuria: 30–300		9 (75.00)		13 (81.25)		22 (78.57)
Macro albuminuria: >300		3 (25.00)		2 (12.50)		5 (17.86)
*Retinopathy*	158.62 ± 140.295	5 (71.43)	136.15 ± 178.691	2 (28.57)	152.20 ± 136.247	7 (6.80)
Optimal: <30		2 (40.00)		1 (50.00)		3 (42.86)
Microalbuminuria: 30–300		2 (40.00)		1 (50.00)		3 (42.86)
Macro albuminuria: >300		1 (20.00)		0 (00.00)		1 (14.28)
**Urinary Creatinine (mg/ dL)**						
*Not specified*	194.46 ± 139.83		130.87 ± 117.85		160.17 ± 131.70	102 (99.03)
Target 20–320 mg/ dL		41 (87.23)		50 (90.91)		91 (89.22)
Non-Target >320 mg/ dL		6 (12.77)		4 (7.27)		10 (9.80)
*CAN*	236.15 ± 150.39	31 (45.59)	123.28 ± 107.24	37 (54.41)	174.74 ± 139.68	68 (66.02)
Target 20–320 mg/ dL		25 (80.65)		34 (91.89)		59 (86.76)
Non-Target >320 mg/ dL		6 (19.35)		2 (5.41)		8 (11.76)
*DPN*	236.84 ± 160.20	21 (47.73)	157.52 ± 149.63	23 (52.27)	195.34 ± 158.11	44 (42.72)
Target 20–320 mg/ dL		17 (80.95)		20 (86.96)		37 (84.09)
Non-Target >320 mg/ dL		4 (19.05)		3 (13.04)		7 (15.91)
*Nephropathy*	256.43 ± 205.44	12 (42.86)	152.65 ± 77.99	16 (57.14)	197.13 ± 152.68	28 (27.18)
Target 20–320 mg/ dL		9 (75.00)		16 (100.0)		25 (89.29)
Non-Target >320 mg/ dL		3 (25.00)		0 (0.00)		3 (10.71)
*Retinopathy*	211.36 ± 55.58	5 (71.43)	159.95 ± 135.98	2 (28.57)	196.67 ± 75.96	7 (6.80)
Target 20–320 mg/ dL		5 (100.0)		2 (100.0)		7 (100.0)
Non-Target >320 mg/ dL		0 (0.00)		0 (0.00)		0 (0.00)
**Albumin–Creatinine Ratio (mg/mmol)**						
*Not Specified*	32.09 ± 52.45	47 (46.08)	39.28 ± 74.58	55 (53.92)	35.97 ± 65.11	102 (99.03)
Optimal: <3		12 (25.53)		10 (18.18)		22 (21.57)
Borderline high: 3–30		23 (48.94)		29 (52.73)		52 (50.98)
High: >30		12 (25.53)		16 (29.09)		28 (27.45)
*CAN*	44.35 ± 60.99	31 (45.59)	45.36 ± 86.19	37 (54.41)	44.90 ± 75.22	68 (66.02)
Optimal: <3		7 (22.58)		6 (16.22)		13 (19.12)
Borderline high: 3–30		12 (38.71)		20 (54.05)		32 (47.06)
High: >30		12 (38.71)		11 (29.73)		23 (33.82)
*DPN*	60.73 ± 68.22	21 (47.73)	35.97 ± 51.28	23 (52.27)	47.79 ± 60.55	44 (42.72)
Optimal: <3		6 (28.57)		5 (21.74)		11 (25.00)
Borderline high: 3–30		4 (19.05)		11 (47.83)		15 (34.09)
High: >30		11 (52.38)		7 (30.43)		18 (40.91)
*Nephropathy*	105.960 ± 57.952	12 (42.86)	111.404 ± 109.675	16 (57.14)	109.071 ± 89.771	28 (27.18)
Optimal: <3		0 (0.00)		0 (0.00)		0 (0.00)
Borderline high: 3–30		0 (0.00)		0 (0.00)		0 (0.00)
High: >30		12 (100.0)		16 (100.0)		28 (100.0)
*Retinopathy*	86.567 ± 87.999	5 (71.43)	58.923 ± 61.616	2 (28.57)	78.671 ± 77.312	7 (6.80)
Optimal: <3		0 (0.00)		0 (0.00)		0 (0.00)
Borderline high: 3–30		2 (40.00)		1 (50.00)		3 (42.86)
High: >30		3 (60.00)		1 (50.00)		4 (57.14)

**Table 5 jcm-11-00903-t005:** Comparison between two machine learning models for each test.

Tests		CAN (pCAN vs. nCAN)	DPN (pDPN vs. nDPN)	RET (pRET vs. nRET)
logistic regression	Accuracy, %	80	55.56	88.54
	Sensitivity, %	85.71	55.77	93.33
	Specificity, %	85.71	55.26	16.67
SVM	Accuracy, %	77.33	67.8	80.5
	Sensitivity, %	29.41	68.89	96.05
	Specificity, %	91.34	66.67	20
RF	Accuracy, %	98.67	67.8	84.38
	Sensitivity, %	100	68.09	97.44
	Specificity, %	98.48	67.44	27.78

**Table 6 jcm-11-00903-t006:** 95% Confidence intervals for cardiac autonomic neuropathy patients (categorical features, such as gender, smoking history, and smokeless tobacco history, have been omited from the table);. The subject count is 75 (65 pCAN, and 10 nCAN patients), and the features that are used in the model classifier are marked in bold text.

Features	Mean	95% CI (Lower Limit to Upper Limit)	
‘Age’	56.167	54.315	58.018
‘Waist Circumference’	141.382	136.503	146.262
‘Diabetes Duration’	15.844	14.571	17.117
‘BMI’	26.657	25.694	27.621
‘Systolic BP’	138.900	134.847	142.953
‘**Diastolic BP**’	**77.600**	**75.355**	**79.845**
‘Weight’	65.517	63.634	67.400
‘Height’	157.399	155.252	159.546
‘HbA1c’	8.799	8.465	9.133
‘Microalbuminuria’	55.741	36.539	74.943
‘Urinary Creatinine’	160.656	131.724	189.588
‘**Albumin–Creatinine Ratio**’	**37.387**	**23.178**	**51.595**

**Table 7 jcm-11-00903-t007:** 95% confidence intervals for diabetic peripheral neuropathy patients (categorical features, such as gender, smoking history, and smokeless tobacco history, have been omited from the table). The subject count is 90 (44 pDPN, and 46 nDPN patients), and the features that are used in the model classifier are marked in bold text.

Features	Mean	95% CI (Lower Limit to Upper Limit)	
‘Age’	55.844	54.017	57.671
‘Waist Circumference’	140.642	135.853	145.432
‘Diabetes Duration’	15.781	14.580	16.983
‘BMI’	26.657	25.713	27.600
‘**Systolic BP**’	**138.385**	**134.449**	**142.322**
‘Diastolic BP’	77.615	75.473	79.756
‘Weight’	65.658	63.824	67.491
‘Height’	157.603	155.475	159.730
‘**HbA1c**’	**8.902**	**8.554**	**9.251**
‘**Microalbuminuria**’	**55.269**	**36.692**	**73.847**
‘Urinary Creatinine’	160.770	133.508	188.032
‘**Albumin-Creatinine Ratio**’	**36.744**	**23.202**	**50.286**

**Table 8 jcm-11-00903-t008:** 95% confidence interval for diabetic retinopathy patients (categorical features, such as gender, smoking history, and smokeless tobacco history, have been omited from the table). The subject count is 96 (7 pRet and 89 nRet patients) and the features that are used in the model classifier are marked in bold text.

Features	Mean	95% CI (Lower Limit to Upper Limit)	
‘Age’	55.707	53.651	57.763
‘Waist Circumference’	139.958	134.525	145.391
‘Diabetes Duration’	15.827	14.486	17.167
‘BMI’	26.817	25.683	27.951
‘Systolic BP’	138.813	134.153	143.474
‘Diastolic BP’	77.347	74.802	79.891
‘**Weight**’	**65.601**	**63.476**	**67.727**
‘Height’	157.177	154.610	159.744
‘**HbA1c**’	**8.955**	**8.553**	**9.356**
‘**Microalbuminuria**’	**67.648**	**44.636**	**90.660**
‘**Urinary Creatinine**’	**172.120**	**139.197**	**205.044**
‘**Albumin-Creatinine Ratio**’	**44.128**	**27.134**	**61.123**

## Data Availability

Data and models could be shared under research agreement with any other researchers working in non-profit organizations.

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
