# Peer review of "Machine Learning for Screening Microvascular Complications in Type 2 Diabetic Patients Using Demographic, Clinical, and Laboratory Profiles"

_jcm, 2022, doi:10.3390/jcm11040903_

Round 1
Reviewer 1 Report
Rashid et al. report on a Bangladesh study that aims to investigate the ability of different machine learning approaches to screen for microvascular complications among people with diabetes. They focus on predictors of microvascular diseases that are easily measured and therefore could be applied in settings with scarce medical resources. Considering the global rise in prevalence of diabetes, this topic is certainly of relevance. However, I am afraid that the study conducted by Rashid et al. is subject to several limitations and might not be suited to advance current knowledge or clinical practice. Below I point out my concerns in more detail.
- The manuscript should be revised regarding the English language.
- In my point of view, the manuscript does not explain how the results of the study should be applied in clinical practice. As it stands, it mainly provides measures on diagnostic test performance, but no guidance on how to use the results for screening (e.g. an algorithm or a simple formula to easily estimate the probability of different microvascular complications in a clinical setting).
- The authors should explain what the benefits of the investigated machine learning methods are compared to simpler approaches, such as logistic regression.
- For diabetic nephropathy, the authors used laboratory measures as predictors/features that were also used to determine the presence of nephropathy. This is not a realistic setting in clinical practice. If the information on the presence of a disease is available from the laboratory measure itself, there is no need to include this laboratory measure in a machine learning model to obtain the same information.
- The sample size is quite small (n = 96) and it is unclear how the desired sample size was determined.
- The authors should provide some information on the representativeness of the study sample.
- No information on the calibration of the models is provided.
- Some of the terms and procedures used in this study need further clarification:
- Please explain in more detail what a multiclass analysis is (ll. 162)
- Tables 1 and 2: It is not totally clear how the numbers were derived. Please provide table legends explaining the abbreviations used in the tables and how the numbers in table 2 add up to the numbers in table 1.
- Section 2.5.2 describes feature selection based on chi-square tests with p < 0.05 as selection criterion. Please explain why the features were selected before implementing the machine learning methods.
- The results should be reported with some measure of statistical uncertainty (e.g. confidence intervals).
- The structure of manuscript confused me at several occasions. In particular, large parts of the methods section (e.g. tables 1 to 3, descriptions of the study population) might be better suited for the results section.
Author Response
Dear Reviewer,
We appreciate your reconsideration of our manuscript in its revised form. The revised manuscript ensure now a better readability to the readers.
Thank you.
regards,
Mohanad

Reviewer 2 Report
I feel this was very well done. You have a done a concise and complete overview of the problem and how machine learning and the algorithms contained with in make a well done final judgement on the predictive value of the biomarkers. Very concise table and graphs which were excellent and well planned. Well done I thoroughly enjoyed reading this paper.
Author Response
Dear Reviewer,
We appreciate your notes and consideration of our manuscript.
Thank you.
regards,
Mohanad
Reviewer 3 Report
Firstly, I would like to suggest you to add the results of logistic regression as a traditional statistical approach. Secondly, I would like to suggest you to compare random forest variable importance findings for binary- vs. 3-class cases.
Author Response
Dear Reviewer,
Thank you for consideration of our work. We have revised the manuscript accordingly.
Editing in manuscript: Logistic regression methods are included in this manuscript and compared with other machine learning models, and showed machine learning provides better result over logistic regression. We have concluded the features showing importance in multiclass classification were used in the binary class classification test.
Thank you.
regards,
Mohanad
Round 2
Reviewer 1 Report
Dear authors,
Thanks very much for the revision of the manuscript. Although some concerns were adequately adressed, there still remain issues, which, in my opinion, are not suffciently adressed. Therefore, I am afraid that the manuscript has not been sufficiently improved to warrant publication. See my comments below for details.
----------------------------------------------
Concern #4: For diabetic nephropathy, the authors used laboratory measures as predictors/features that were also used to determine the presence of nephropathy. This is not a realistic setting in clinical practice. If the information on the presence of a disease is available from the laboratory measure itself, there is no need to include this laboratory measure in a machine learning model to obtain the same information.
Author’s Response: We have used microalbuminuria as predictor to classify diabetic nephropathy and the result is included in the manuscript. Like other types of complication of T2DM in this study, the presence of nephropathy (as determined by DCL profiles) can be investigated through machine learning to provide an indicator which patients need to be treated with a single or multiple complications. As nephropathy in diabetics is a critical condition that can result in significant morbidity and/or mortality, determining its presence along with other complications can lead to optimizing therapy with expected better outcome.
Reviewer’s Response: I agree that nephropathy is an important complication of diabetes. However, I still don’t think it is useful to use the albumin-creatinine ratio as a predictor in a prediction model for nephropathy when the albumin-creatinine ratio was also used to diagnose nephropathy, as is described in the ll. 155 of the manuscript. In that case, nephropathy status can be determined directly from albumin-creatinine ratio instead predicting it with a prediction model.
---------------------------------------------
Concern #5: The sample size is quite small (n = 96) and it is unclear how the desired sample size was determined.
Response: Although the sample size included in this study was slightly small, we collected a lot of patient clinical information. In machine learning, training models using a lot of variables balances the small sample size. In addition, we tested the algorithm using a leave-one-out training/testing scheme, which ensures the inclusion of the maximum number of subjects within the trained model. Therefore, a better generalizability can be achieved.
Reviewer’s Response: I don’t think that using a lot of variables balances the disadvantages of small sample size. In contrast, a small sample size usually implies that only a small number of variables can be assessed simultaneously in a prediction model.
---------------------------------------------
Concern #9: The results should be reported with some measure of statistical uncertainty (e.g. confidence intervals).
Response: Due to the lower sample size, statistical uncertainty measurement doesn’t reflect well in this specific study.
Reviewer’s Response: I don’t think that this is a valid point. In contrast, particularly with small sizes, it is important to quantify statistical uncertainty, because it is usually substantial in these cases.
Author Response
Dear Reviewer,
Thank you for re-considering our manuscript in its revised form. We hope that the current version satisfies all the concerns and has improved in terms of the readability to the readers.
Please find our responses below,
----------------------------------------------
Concern #4: For diabetic nephropathy, the authors used laboratory measures as predictors/features that were also used to determine the presence of nephropathy. This is not a realistic setting in clinical practice. If the information on the presence of a disease is available from the laboratory measure itself, there is no need to include this laboratory measure in a machine learning model to obtain the same information.
Author’s Response #1: We have used microalbuminuria as predictor to classify diabetic nephropathy and the result is included in the manuscript. Like other types of complication of T2DM in this study, the presence of nephropathy (as determined by DCL profiles) can be investigated through machine learning to provide an indicator which patients need to be treated with a single or multiple complications. As nephropathy in diabetics is a critical condition that can result in significant morbidity and/or mortality, determining its presence along with other complications can lead to optimizing therapy with expected better outcome.
Reviewer’s Response: I agree that nephropathy is an important complication of diabetes. However, I still don’t think it is useful to use the albumin-creatinine ratio as a predictor in a prediction model for nephropathy when the albumin-creatinine ratio was also used to diagnose nephropathy, as is described in the ll. 155 of the manuscript. In that case, nephropathy status can be determined directly from albumin-creatinine ratio instead predicting it with a prediction model.
Author’s Response #2: The authors appreciate the concern raised by this reviewer. We do agree that albumin-creatinine is the commonly and optimum metric used to diagnose nephropathy. Accordingly, we have excluded this feature and re-trained the model using the second highly important feature determined by the Chi-squared test (microalbuminuria, See Fig. 5, page 18). More information was added to the manuscript accordingly (See Section 3.3.3).
----------------------------------------------
Concern #5: The sample size is quite small (n = 96) and it is unclear how the desired sample size was determined.
Author’s Response #1: Although the sample size included in this study was slightly small, we collected a lot of patient clinical information. In machine learning, training models using a lot of variables balances the small sample size. In addition, we tested the algorithm using a leave-one-out training/testing scheme, which ensures the inclusion of the maximum number of subjects within the trained model. Therefore, a better generalizability can be achieved.
Reviewer’s Response: I don’t think that using a lot of variables balances the disadvantages of small sample size. In contrast, a small sample size usually implies that only a small number of variables can be assessed simultaneously in a prediction model.
Author’s Response #2: The authors agree with the reviewer that the sample size used in this study is small. We have added this information as a limitation for the current study (See Section 5).
“It is worth noting that the proposed study should be further validated on a wider patient cohort to strengthen the observations. Although the findings of this study were promising and correlate with the observations found in literature, one limitation to the current work was the small sample size, which is a common situation in biomedical studies that rely on patient data. Overall, RF and SVM are known to handle small sample sizes with high performance capabilities [71,72], especially when compared to other artificial intelligence algorithms such as deep neural networks that require big data. Therefore, an essential future direction to the current study is to be tested on big clinical data and with additional machine/deep learning algorithms.”
We would like to mention that the current study has ended, and it is not possible to obtain more patient data. In addition, the current study provides more details on the performance of computerized algorithms in diabetes research and is not considered as a prevalence study.
---------------------------------------------
Concern #9: The results should be reported with some measure of statistical uncertainty (e.g. confidence intervals).
Author’s Response #1 : Due to the lower sample size, statistical uncertainty measurement doesn’t reflect well in this specific study.
Reviewer’s Response: I don’t think that this is a valid point. In contrast, particularly with small sizes, it is important to quantify statistical uncertainty, because it is usually substantial in these cases.
Author’s Response #2: Based on the concern raised by this reviewer, we have added 95% confidence interval as a measure of statistical uncertainty for each test (CAN, DPN, NEP, and RET) and added tables accordingly to show them in the manuscript (See Section 3.3, See Table 6, 7, 8, and 9).
---------------------------------------------
Regards,
Mohanad